

# Inflammatory auxo-action in the stem cell division theory of cancer

Yi Luo[1,2] and Jian-Hui Xiao[1,2,3]

[1] Institute of Medicinal Biotechnology, Affiliated Hospital of Zunyi Medical University, Zunyi, China
[2] Zunyi Municipal Key Laboratory of Medicinal Biotechnology & Guizhou Provincial Research Center for Translational Medicine, Affiliated Hospital of Zunyi Medical University, Zunyi, China
[3] Department of Gynaecology and Obstetrics, Affiliated Hospital of Zunyi Medical University, Zunyi, China

## ABSTRACT

Acute inflammation is a beneficial response to the changes caused by pathogens or injuries that can eliminate the source of damage and restore homeostasis in damaged tissues. However, chronic inflammation causes malignant transformation and carcinogenic effects of cells through continuous exposure to pro-inflammatory cytokines and activation of inflammatory signaling pathways. According to the theory of stem cell division, the essential properties of stem cells, including long life span and self-renewal, make them vulnerable to accumulating genetic changes that can lead to cancer. Inflammation drives quiescent stem cells to enter the cell cycle and perform tissue repair functions. However, as cancer likely originates from DNA mutations that accumulate over time via normal stem cell division, inflammation may promote cancer development, even before the stem cells become cancerous. Numerous studies have reported that the mechanisms of inflammation in cancer formation and metastasis are diverse and complex; however, few studies have reviewed how inflammation affects cancer formation from the stem cell source. Based on the stem cell division theory of cancer, this review summarizes how inflammation affects normal stem cells, cancer stem cells, and cancer cells. We conclude that chronic inflammation leads to persistent stem cells activation, which can accumulate DNA damage and ultimately promote cancer. Additionally, inflammation not only facilitates the progression of stem cells into cancer cells, but also plays a positive role in cancer metastasis.

## INTRODUCTION

Cancer is a major reason for decreasing life expectancy (*Bray et al., 2021*). Cancer-related mortality remains one of the leading causes of death globally, accounting for 13% of all human deaths despite the availability of a range of clinical treatment strategies (*Bray et al., 2018*). The term ''cancer'' is fearsome to people because it is rooted in misunderstandings or misconceptions. For a more accurate understanding of cancer, our forefathers (as early as ancient Greece) actively explored its origin and proposed humoral theories, which stated that the imbalance of humors, such as black bile, was responsible for various diseases (*Karpozilos & Pavlidis, 2004*). With the development of modern science and technology, scientists have presented many different theories for the origin of cancer, including

Corresponding author
Jian-Hui Xiao, jianhuixiao@126.com

field theory, chemical carcinogenesis, infection, chromosomal abnormalities, mutations, epigenetic changes, non-healing wounds, and immune surveillance theories, over the past century (*Liu, 2018*; *Allegra et al., 2014*; *Sell, 2010*). The "monoclonal theory," proposed by Hanahan and Weinberg in 2000 remains the mainstream and widely accepted theory of tumor development mechanisms (*Hanahan, 2000*). It states that tumors originate from a single "rebellious" cell, which eventually grows into the whole tumor. Findings based on this theory have exerted positive effects on cancer prevention and therapy by reducing incidence and mortality. However, it does not provide a complete framework for understanding the cellular origin and pathogenesis of many cancers, which is necessary for providing reliable roadmaps for cancer prevention, diagnosis, prognosis, and treatment (*Pandya et al., 2021*). Cancer is known to be ultimately caused by uncontrolled cell growth and proliferation. This feature is similar to the self-renewal ability of stem cells, which is their exclusive potential to generate an unlimited number of cells. Although the current view is that chemotherapy kills most cells in cancer tissues, cancer stem cells (CSCs) are believed to be left behind, which may cause recurrence (*Dean, Fojo & Bates, 2005*). Furthermore, cancer cells possessing unique stem cell-like properties have been widely identified in different human cancers over the last two decades (*Zhou et al., 2021*; *Paul, Dorsey & Fan, 2022*). Since the beginning of the new century, the CSC theory has been the focus of an upsurge in research (*Hung, Yang & Kao, 2019*). Nevertheless, the origin of these cancer cells remains controversial. Currently, numerous data point to resident adult stem cells (ASCs) or primitive progenitor cells as the origin of cancer cells, and studies have emphasized their role not only as propagators of repair after tissue damage but also as cancer initiators (*White & Lowry, 2015*). Other studies have suggested that the accumulation of mutations from stem cell division is the main trigger of cancer, and there is a strong correlation between cancer risk and stem cell division (*Tomasetti & Vogelstein, 2015*; *Tomasetti, Li & Vogelstein, 2017*). In 2015, based on fundamental concepts in cell biology and related evidence, Spanish scientist Lopez-Lazaro proposed the stem cell division theory of cancer (*López-Lázaro, 2015*; *López-Lázaro, 2018*).

Stem cells in developing embryos can proliferate indefinitely or remain undifferentiated. When an organism needs to repair or remodel a certain tissue, the stem cell genome is activated in a designated manner to differentiate into somatic cells with specialized functions and identities (*Miller & Kaplan, 2012*). ASCs with similar functions remain in individuals after embryonic development (Fig. 1). It is generally accepted that ASCs naturally exist in a quiescent state, with a small cell size and no cell division (*Cheung & Rando, 2013*). Quiescent stem cells can be "awakened" into the cell cycle in response to local damage signals or other regeneration needs (*Machado et al., 2021*), and mutations leading to cancer tend to occur only in actively dividing cells (*Zhu et al., 2016*). Nevertheless, cell division is a major source of DNA alterations in normal stem cells and can lead to the accumulation of various cancer-promoting errors. Recent cancer statistics show a dramatic increase in cancer incidence with age, suggesting that the formation of most cancers requires multistep accumulation of deoxyribonucleic acid (DNA) changes over years or decades (https://seer.cancer.gov/csr/1975_2018/). Accordingly, the researchers also found that stem cells from newborn mice were less likely to become cancerous than those from
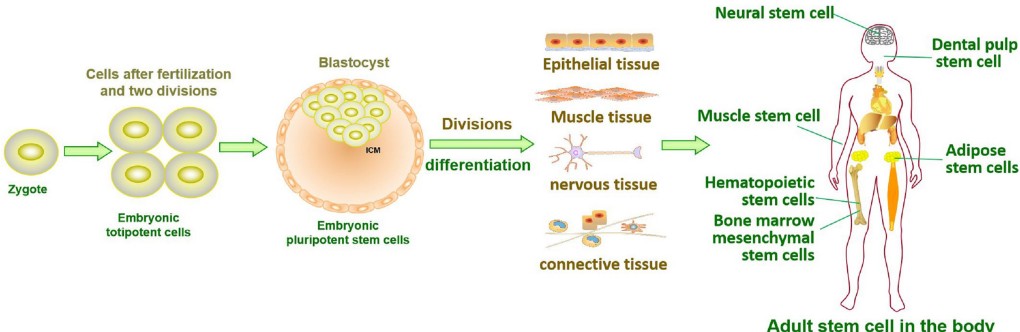

**Figure 1  Tissue formation and distribution of adult stem cells in the body.**

adult mice (*Zhu et al., 2016*). In the past, DNA damage was considered to result only in genomic instability. However, recent evidence has shown that DNA damage can trigger inflammation by activating the cGAS-STING axis and activating NF-$\kappa$B through ATM or ATR proteins (*Zhao et al., 2023*). Furthermore, DNA damage induces the expression of type I interferon (IFN) and other inflammatory cytokines(*Brzostek-Racine et al., 2011*; *Kondo et al., 2013*; *Härtlova et al., 2015*). Thus, the DNA damage generated during stem cell division acts on itself to stimulate abnormal division by activating inflammatory pathways or inducing the production of inflammatory cytokines. Alternatively, inflammation induced by tissue damage can "wake up" quiescent stem cells and cause them to divide, greatly increasing the risk of cancer (*Nichane et al., 2017*; *Qi et al., 2018*). Overall, a circulating network of inflammatory signals produced by multiple pathways regulates the process of stem cell division after tissue damage (Fig. 2). Combined with the stem cell division theory of cancer (*López-Lázaro, 2015*; *López-Lázaro, 2018*), we can conclude that tissue damage creates favorable conditions for the formation of cancer cells, and inflammation inadvertently promotes cancer (*Nolan et al., 2022*). In addition, while many factors can increase cancer risk, including smoking, alcohol, high body mass index, pathogens, and radiation, the abnormal activation of inflammatory signals is a common underlying factor associated with all these risk factors (*GBD 2019 Cancer Risk Factors Collaborators, 2022*). Therefore, investigating the effects of inflammation on the behavior of normal stem cells, CSCs, and cancer cells based on the stem cell division theory of cancer is critical to understanding cancer generation from multiple perspectives. The information we have gathered and reviewed here will be of interest to researchers working in developmental biology and cancer development, as well as those working in the fields of molecular biology, cell biology, and stem cells.

## SURVEY METHODOLOGY

The PubMed (https://pubmed.ncbi.nlm.nih.gov/) and Google Scholar (https://scholar.google.com/) repositories were used to search the following terms: stem cells, cancer stem cells, progenitor cells, mesenchymal stem cells, cancer cells, cancer, inflammation,

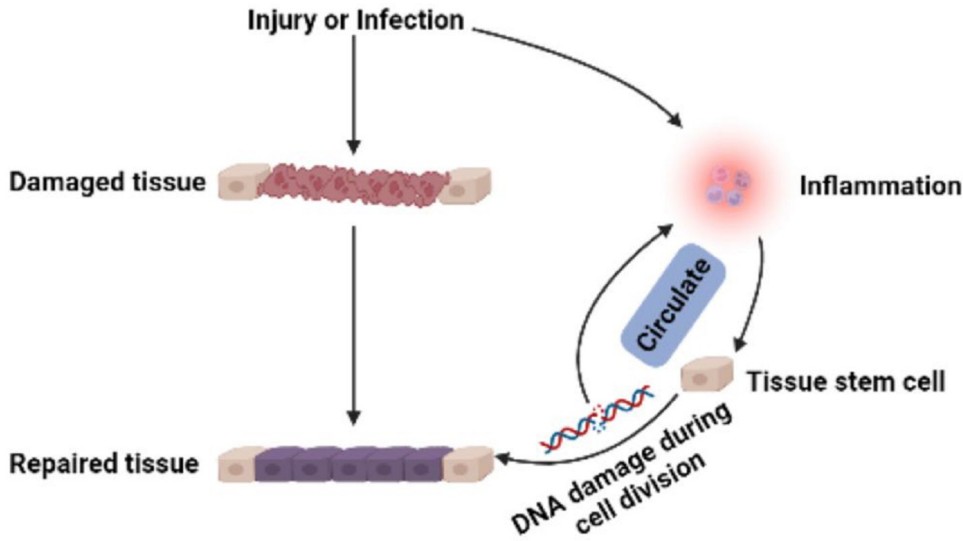

**Figure 2 Multi-pathway inflammatory signaling and tissue repair.**

cytokines, cell division, malignant phenotype, tumorigenicity, cancer metastasis, stemness, pluripotency, self-renewal, and differentiation potential.

## Stem cell division theory of cancer

Cancer remains one of the major diseases that threaten human survival and is difficult to cure clinically (*Bray et al., 2018*). Despite significant progress in understanding the signals that drive cancer growth and how to target these signals, effective disease control remains a key scientific and medical challenge. Three well-known reasons behind the difficulty in conquering cancer include tumor immune escape (difficulty of the host immune system in recognizing cancer cells), cancer metastasis, and tumor resistance. Although cancer cells have different biochemical compositions, antigenic structures, and biological behaviors than normal cells (*Liebelt, Finocchiaro & Heimberger, 2016*; *Pardoll, 2015*), the immune system struggles to identify cancer cells accurately. Tracing the origin of cancer cells is essential for finding strategies to overcome this phenomenon. Cancer cells are mutated from "self" cells, which endows them with good camouflage properties. Cancer metastasis is the other main reason for the failure of clinical treatment of tumors and death of most patients. Although clinical manifestations of cancer suggest that metastasis is a late event, many recent studies have demonstrated that cancer cells begin to metastasize at an early stage, even before diagnosis (*Hosseini et al., 2016*; *Hu et al., 2020*; *Harper et al., 2016*). Researchers analyzed samples from more than 100 patients with breast, colorectal, and lung cancer and found that the genomic drivers required for invasion and metastasis were present in primary tumors (*Hu et al., 2020*). Breast cancer cells with certain molecular changes can spread to other organs even before the primary tumor forms. These cells remain quiescent for long periods before being awakened to form aggressive and lethal metastatic breast cancer cells (*Harper et al., 2016*). This new model of early metastasis challenges our understanding of how cancer spreads and forms metastases. Therefore, the
novel biological mechanisms of early cancer spread must be explored to target metastatic cancer cells. A key and well-recognized factor that promotes cancer metastasis is the formation of a microenvironment at a specific site (*Bado et al., 2021*) that is favorable for cancer metastasis, including tumor-secreted factors, mobilization of suppressive immune cells, and inflammatory polarization of stromal components at this site (*Liu & Cao, 2016*). During the multistep process of cancer metastasis, primary cancer cells acquire cellular and phenotypic plasticity to survive and grow in diverse microenvironments (*Zhang et al., 2021*). Therefore, understanding the novel biological mechanisms of cancer metastasis, cancer cell origin, and tumor microenvironment is essential.

Although the target cells of the transforming mutation are unknown for most cancers, recent evidence suggests that cancer is a stem cell-based disease, and embryonic, adult, or pluripotent stem cells are the origin cells (*Sánchez-Danés et al., 2016*). Research shows that the number of stem cell divisions in tissues is positively correlated with cancer incidence, and 65% of cancers can be explained by this number (*Quesenberry & Goldberg, 2015*). Therefore, *López-Lázaro (2015)* proposed a theory of stem cell division in cancer, which states that (1) tumors originate from normal stem cells; (2) the main determinant of cancer development is the damage undergone by stem cells during division; (3) the accumulation of sufficient damage in stem cells leads to the production of tumor stem cells, which are responsible for tumor formation; and (4) metastasis occurs when stem cells (uncontrolled, precancerous, or cancerous) or their malignant progeny leave their natural tissue (not necessarily the primary tumor) and form tumors elsewhere.

The stem cell division theory of cancer is controversial; however, the idea that cancer originates from normal stem cells has a solid biological basis. Several observational studies have shown that age is the most important risk factor for cancer (*Zinger, Cho & Ben-Yehuda, 2017*; *Hsu, 2016*). Stem cells possess genome passed on from fertilized eggs, and remain in our body until death; therefore, they are the only cells that acquire and accumulate DNA changes throughout our lives (*ILW, 2000*). Self-renewal, which is the most important property of stem cells, is strikingly similar to that of cancer cells. Many canonical signaling pathways associated with cancer, such as Notch, sonic hedgehog, and Wnt signaling, also regulate normal stem cell development. Emerging cancer cells likely take advantage of the self-renewal cell division machinery normally expressed in stem cells. Understanding the signaling network of normal stem cells and cancer cell development will help to trace the initial cancer cells and identify targets for anticancer therapy. Stem cell functions, including splitting rate and migration ability, are established within the niche (*Ge et al., 2020*; *Duckworth, 2021*; *Li & Guo, 2021*). The stem cell niche is a dynamic and special microenvironment that activates certain intracellular signaling pathways by producing factors that directly act on stem cells, ultimately determining the fate of cells, such as division, differentiation, or apoptosis. Additionally, tissue stem cells, which are usually in a quiescent state, activate self-renewal and differentiation programs to maintain tissue homeostasis and repair wounds (*Hsu & Fuchs, 2012*). Tissue damage accompanies the generation of sterile and pathogenic inflammation (*Zindel & Kubes, 2020*). Thus, inflammatory signals regulate stem cell function in many ways and undoubtedly affect the accumulation of stem cell mutations.

### Is the inflammatory response a friend or foe to cancer?

The typical triggers of inflammation (infection and tissue damage) are marked by leukocyte infiltration, a process by which the innate immune system helps repair damaged tissue by activating immune and non-immune cells against pathogens (*Sun, 2017*). Currently, the medical community has made it clear that inflammation is closely related to cancer occurrence, development, and efficacy of anticancer treatment and acts as a "double-edged sword". Inflammation can be classified into acute and chronic inflammation. Acute inflammation is the natural defense of the body against damaged cells, viruses, and other harmful stimuli; it sets in quickly and helps the body to heal itself. During this process, the immune cells and chemicals involved can kill pathogens, promote tissue repair, prevent tumor growth, and activate the immune system through stimuli and inflammatory factors, thereby acting as tumor suppressors (*Castaño et al., 2018*; *Rossi, Jamieson & Weissman, 2008*; *Schreiber, Old & Smyth, 2011*). For example, induced acute inflammation is used to treat bladder cancer (*Askel et al., 2012*), and high concentrations of tumor necrosis factor (TNF) can induce an antitumor response in a mouse model of sarcoma (*Havell, Fiers & North, 1988*). Acute inflammation is short-lived and quickly resolved. If inflammation cannot be resolved in time, it results in chronic inflammation, which promotes the development of malignant tumors through continuous exposure to pro-inflammatory factors and activation of signaling pathways, such as NF-$\kappa$B and signal transducer and activator of transcription (STAT) 3. When cells become malignant, chronic inflammation signals play a critical role in promoting cancer cell proliferation, evading immune surveillance, and promoting angiogenesis to support the growth and spread of cancer. For example, low and sustained levels of TNF-$\alpha$ can induce tumor phenotypes, and interleukin (IL) 1 $\beta$ has been identified as a key molecule contributing to colorectal cancer (CRC) development (*Balkwill, 2006*; *Muthusami et al., 2021*). IL-6, a pro-inflammatory cytokine with typical pro-cancer effects, plays a key role in promoting proliferation and inhibiting apoptosis by binding to its receptor IL-6Ra and co-receptor glycoprotein130 (gp130) to activate transcription factors, including STAT1 and STAT3 (*Hirano, 2021*).

Inflammation is an adaptive response triggered by infection and tissue damage and is differentiated by pathogen-associated molecular patterns (PAMPs) and damage-associated molecular patterns (DAMPs). PAMPs, such as membrane-associated lipids and lipopolysaccharides, are exogenous components specific to invading microorganisms. In contrast, inflammation without pathogens and their products is called sterile inflammation and is triggered by endogenous danger signs. DAMPs are released upon tissue injury (*Medzhitov, 2008*) and initial inflammation and leukocyte recruitment are prerequisites for effective tissue repair (*McDonald et al., 2010*). Researchers have now found that both PAMPs and DAMPs are recognized by pattern recognition receptors (PRRs) to initiate immune responses. These receptors include Toll-like receptors (TLRs), RIG-i-like receptors (RLRs), C-type lectin receptors (CLRs), and Nod-like receptors (NLRs), which induce cytokine and IFN production by activating associated inflammatory pathways (*Wicherska-Pawłowska, Wróbel & Rybka, 2021*). After identifying PAMP and DAMP, TLRs bind to adaptor proteins containing Toll-IL-1-resistance (TIR) domains, such as myeloid differentiation primary response gene 88 (MYD88) or Toll-receptor-associated

activator of interferon (TRIF). MyD88 and TRIF recruit and activate mitogen-activated protein kinases (MAPKs) and I$\kappa$B kinases (IKK) and ultimately induce the expression of inflammatory cytokines by activating transcription factor activating protein 1 (AP-1) and NF-$\kappa$B, respectively (*O'Neill, Golenbock & Bowie, 2013*; *Gong et al., 2020*). Thus, the production of PAMPs and DAMPs is necessary for tissue repair and regeneration; however, under persistent inflammation, they can also lead to the development of cancer (*Gong et al., 2020*). For example, pro-inflammatory microenvironments mediated by the TLR2-MyD88-NF-$\kappa$B pathway support epithelial ovarian CSCs-driven repair and self-renewal (*Chefetz et al., 2013*).

## Influence of inflammation on non-malignant stem cell behavior

ASCs can undergo extensive cell division while retaining the capacity to generate stem cells and differentiate into specialized cells.This enables ASCs to replensh damaged tissues and help maintain healthy tissue in the body throughout an organism's lifespan (*Goodell, Nguyen & Shroyer, 2015*). ASCs can remain quiescent without dividing for extended periods until they are activated (*Sueda & Kageyama, 2020*). Accordingly, ASCs are suspected to be the origin of cancer cells because of their long lifespan and propensity for extensive cell division (*Rossi et al., 2020*).

### Inflammatory response: the key to awakening quiescent stem cells

Historically, ASCs have been considered to exist in either a quiescent state, in which the cell is not actively cycling, or an activated state, in which the cell has entered the cell cycle (*Cheung & Rando, 2013*; *Rossi et al., 2012*). In the quiescent state, stem cells are in the reversible G0 phase and can re-enter the cell cycle in response to normal physiological stimuli. This is different from the G0 phase for terminally differentiated cells or senescent cells, which have left cell cycle and cannot re-enter (*Cheung & Rando, 2013*). The non-proliferative quiescent state of ASCs can protect the cell's DNA from mutations acquired during continuous cell division (*Walter et al., 2015*), making them less likely to acquire cancer.

Many factors can trigger ASCs to exit the quiescent state. ASCs activation can be induced during tissue damage through mechanisms such as cell–cell contact and cell–extracellular matrix interaction (*Cho et al., 2019*). In addition, the mechanism by which inflammation awakens quiescent stem cells to promote the regeneration of damaged tissues has been widely reported (*Schuettpelz & Link, 2013*). Reactive oxygen species (ROS) generated during tissue damage and infection are important cues linking inflammation to intestinal stem cells (ISCs) proliferation through activating Jun N-terminal kinases (JNKs) and inhibiting nuclear factor erythroid2-related factor 2 (Nrf2) signal (*Hochmuth et al., 2011*; *Biteau, Hochmuth & Jasper, 2008*). Lipopolysaccharide (LPS) -induced transient mild inflammation stimulates the selective activation and proliferation of stationary stem cell populations, such as Clara-like cells (CLCs) (*Verckist et al., 2018*). Moreover, IFN-$\gamma$ treatment increased the expression of bone marrow stromal cell antigen 2 (BST2), a cell surface protein essential for INF$\gamma$-dependent hematopoietic stem cell (HSC) activation, thereby disrupting HSC quiescence and promoting excessive terminal differentiation (*Florez et al., 2020*). Additionally, IL-6 and TNF directly act on epithelial cells by binding to

the IL-6 receptor, gp130 heterotetramer, and TNF receptor 1 (TNFR1), thereby promoting the regeneration of damaged intestinal mucosa (*Grivennikov et al., 2009*; *Böhm et al., 2010*). IL-22 acts on epithelial cells and fibroblasts to stimulate proliferation, inhibit cell death, and delay terminal differentiation. The binding of IL-22 to its receptor leads to the activation of Janus kinases (JAK)-STAT3, MAPK, extracellular signal-regulated kinase (ERK), and JNK (*Nikoopour, Bellemore & Singh, 2015*). Therefore, inflammation is an important mediator that mobilizes quiescent stem cells to regenerate after tissue damage.

### Inflammation mobilizes tissue stem cells: regeneration and tumorigenicity coexist

After tissue damage, inflammatory cells and cytokines are the main components of the tissue stem cell niche. Tissue stem cells, which are activated from a quiescent state owing to niche changes, repair damaged tissues through proliferation and differentiation and exhibit strong regenerative capabilities (*Naik et al., 2018*). For example, multiple subsets of innate lymphoid cells can play context-specific roles in stem cell regeneration and differentiation (*Lindemans et al., 2015*; *von Moltke et al., 2016*). However, mutations can occur with cell division. Although stem cells can self-renew for extended periods, but this also poses an inherent challenge. Being the longest-lived cells in an organism, they are at an increased risk of acquiring genomic damage, which undoubtedly increases the likelihood of accumulation of mutations in the stem cell genome. Gradual accumulation of genomic mutations throughout life may lead to carcinogenesis. Therefore, under the influence of inflammation, the regeneration of tissue stem cells coexists with tumorigenic ability. Although mutations and cancer incidence in some tissues depend primarily on exposure to external mutagens (*Yoshida et al., 2020*), intrinsic factors, such as the number of cell divisions in the tissue, appear to dominate other cancer types (*Cheung & Rando, 2013*). For example, a high-fat diet can drive a surge in intestinal stem cells and produce a range of other cells that behave very similarly to stem cells by multiplying indefinitely and differentiating into other types of cells. Stem and stem-like cells are more likely to cause intestinal tumors (*Beyaz et al., 2016*; *Beyaz et al., 2021*). Wnt/ $\beta$-catenin signaling plays a key role in intestinal epithelial homeostasis by promoting stem cell renewal. If Wnt signaling is overstimulated, stem cells divide uncontrollably. Mice with mutations in the Wnt signaling pathway develop polyps that eventually develop into colon cancer (*Degirmenci et al., 2018*).

### Influence of inflammatory response on adult stem cell function

The function of ASCs has been established in the stem cell niche, which regulates stem cell survival and function by producing factors that directly act on stem cells (*Ge et al., 2020*). Inflammatory mediators, which are important niche components after tissue injury, influence the behavior of stem cells in various ways. Numerous studies have reported that the inflammatory microenvironment affects the biological characteristics and behavior of stem cells, both *in vitro* and *in vivo* (Table 1). For example, IL-1-induced mesenchymal stem cells (MSCs) show increased granulocyte colony-stimulating factor (G-CSF) expression through the type 1 IL-1 receptor, which results in decreased secretion of LPS-activated microglia inflammatory mediators, priming MSCs for *in vitro* transformation

to an anti-inflammatory and pro-nutritional phenotype (*Redondo-Castro et al., 2017*). Moreover, a combination of four pro-inflammatory cytokines (IL-1 $\alpha$, IL-13, TNF-$\alpha$, and IFN-$\gamma$) secreted by T cells can promote the continuous expansion of muscle stem cells (MuSCs) *in vitro* for over 20 generations (*Fu et al., 2015*). IL-22 promotes ISC-mediated epithelial regeneration by inducing STAT3 phosphorylation (*Lindemans et al., 2015*). The inflammatory milieu mediated by IL-1 and TNF-$\alpha$ preconditioning initially stimulates the regeneration of gingival mesenchymal stem/progenitor cells (G-MSCs); this positive effect disappears as inflammation persists, followed by a short-term stimulatory effect on osteogenesis (*Zhang et al., 2017*). After exposing cultured MuSCs to the inflammatory mediator prostaglandin E2 (PGE2) for one day, the number of cells increased six-fold compared to the control group (*Ho et al., 2017*). Thus, these studies confirmed that inflammatory factors activate downstream stem cell pathways, enhancing their self-renewal and *in vivo* efficacy. In addition, when MSCs are exposed to an inflammatory environment, they release more functional growth factors, exosomes, and chemokines, thereby regulating the inflammatory microenvironment and promoting tissue regeneration (*Liu et al., 2022*). *In vitro* studies have confirmed that some inflammatory factors have profound effects on stem cell behavior, especially proliferation and immune phenotypes. Considering the stem cell division theory of cancer and the coexistence of stem cell proliferation and tumorigenicity, it is uncertain whether the inflammatory microenvironment, especially the influence of inflammatory factors on stem cell behavior, is beneficial or harmful to cancer formation. However, it is certain that stem cells retain their biological properties, such as self-renewal and tissue repair, after coming into contact with inflammatory factors. Partial inflammatory factors can maintain or even improve stem cell function; that is, inflammatory factor treatment indirectly maintains the integrity of the stem cell genome.

*In vivo* studies have confirmed that the tissue repair function of bone marrow MSCs in various inflammatory diseases depends on inflammation regulation and production of various growth factors (*Shi et al., 2010*; *Wang et al., 2014*). Thus, MSCs localized in damaged tissues can alleviate inflammation and promote tissue stem/progenitor and other resident cells to regenerate normal functioning cells and improve the tissue microenvironment by acting synergistically. For example, *in vivo* ISC turnover is heavily regulated by the crypt niche. Macrophages have been identified as a key component of intestinal crypts, and their production of PGE2 promotes Wnt/$\beta$-catenin signaling by binding to the ISC prostaglandin E receptors (EP, also know as Ptger) 1 and EP4, which in turn promotes their self-renewal (*Zhu et al., 2022*). Furthermore, when mice were injected with PGE2 intramuscularly, muscle regeneration was significantly promoted and muscle strength increased. Conversely, when MuSC responses to PGE2 were inhibited, EP4 expression was reduced, or nonsteroidal anti-inflammatory drugs were administered to suppress PGE2 production, strength recovery was hampered (*Ho et al., 2017*). After intestinal injury, innate lymphocytes produce a large amount of IL-22, which directly targets ISCs, enhances the growth of small intestinal organoids, and promotes ISC proliferation and expansion (*Lindemans et al., 2015*). The interaction between ISCs and T cells in the local microenvironment has been elucidated, suggesting that T helper (Th) 1, Th2, and Th17 cells and their cytokines IFN-$\gamma$, IL-13, IL-17, and other pro-inflammatory signals can promote

**Table 1  Inflammatory soluble factors or immune cells and their effect on stem cell function.**

| Soluble factors or immune cell | Cellular origin | Stem cell type | Effects on stem cell function | References |
|---|---|---|---|---|
| IL-1 | Macrophages | Bone marrow derived MSC | Prime MSCs towards an anti-inflammatory and pro-trophic phenotype in vitro | *Redondo-Castro et al. (2017)* |
| IL-1 $\alpha$/IL-13/TNF- $\alpha$/IFN- $\gamma$ | T cell/ macrophages | Muscle stem cells (MuSCs) | Serially expand MuSCs *in vitro* for over 20 passages | *Fu et al. (2015)* |
| IL-22 | T cell | Intestinal stem cell (ISC) | Augment the growth of mouse and human intestinal organoids, increase ISC proliferation and expansion. | *Lindemans et al. (2015)* |
| PGE2 | Inflammatory cells | ISC | Promote ISC self-renewal | *Zhu et al. (2022)* |
| PGE2 | Inflammatory cells | MuSC | Promote MuSC proliferation | *Ho et al. (2017)* |
| Tregs | T cell | Hair follicle stem cells (HFSCs) | Augment HFSCs proliferation and differentiation | *Ali et al. (2017)* |
| IFN- $\gamma$ / IL-13/ IL-17 | T cell | ISC | Promote ISC differentiation | *Biton et al. (2018)* |
| IL-10 | T cell | ISC | Promote ISC self-renewal | *Biton et al. (2018)* |
| IL-1/TNF- $\alpha$ | Tcell/macrophages | Gingival mesenchymal stem/progenitor cells (G-MSCs) | Stimulate G-MSCs regeneration and osteogenesis | *Zhang et al. (2017)* |

the differentiation of Lgr5+ISCs, while regulatory T cells (Treg) cells and their cytokine IL-10 promote ISC self-renewal (*Biton et al., 2018*). Skin-resident Tregs express high levels of the Notch ligand family member Jagged 1 (Jag1). Jag1 expression on Treg promotes hair follicle regeneration by enhancing hair follicle stem cell proliferation and differentiation (*Ali et al., 2017*). Additionally, impaired communication between aging keratinocytes and immune cells causes difficulty in wound healing in aging skin (*Keyes et al., 2016*). In summary, *in vivo* studies have confirmed the importance of inflammatory signaling in regulating tissue regeneration by promoting the proliferation and differentiation of tissue-resident stem cells. Both *in vitro* and *in vivo* studies have confirmed that the inflammatory environment profoundly affects the biological functions and behavior of stem cells, including enhanced self-renewal and tissue repair. Based on the stem cell division theory of cancer, cancer cells are derived from normal stem cells; therefore, the inflammatory microenvironment is a pivotal factor affecting cancer cell formation from the source. However, under different conditions, the microenvironments of stem cells are complex, and the influence of the inflammatory microenvironment on the function of stem cells and their development into cancer requires further exploration.

## Influence of inflammation on cancer stem cell behavior

CSCs, a small subgroup of malignant tumor cells, have enhanced self-renewal, metastasis, and spread as well as treatment resistance, and play a key role in the occurrence, recurrence and metastasis of tumors (*Huang et al., 2020*). Currently, it is considered that there are many similarities between CSCs and ASCs, especially in their self-renewal, signal pathways and some stemness transcription factors. Many signaling pathways that contribute to the survival, proliferation, self-renewal, and differentiation characteristics of ASCs are abnormally activated or inhibited in CSCs (*Clara et al., 2020*; *Yang et al., 2020*). For example, Wnt signaling is one of the key cascades regulating stem cell development and survival; however, its increased activity induces stem cell signatures in colorectal cancer (CRC) cells (*Zhan, Rindtorff & Boutros, 2017*; *Essex et al., 2019*). The PI3K/Akt/mTOR pathway is also important for stem cell survival, proliferation, and migration, and its continued activation in CSCs induces tumorigenesis, cancer metastasis, and drug resistance (*Karami Fath et al., 2022*). Recently, a mainstream study showed that CSCs originate from ASCs or progenitor cells (*Zhang et al., 2017*), and that many oncogenic factors, including inflammation, induce the formation of CSCs by activating the pathways necessary for them (*Gasmi et al., 2022*).

### Inflammation drives the generation of cancer stem cells

In some tumors, CSCs are considered to be a mutated version of non-malignant stem/progenitor cells and a precursor to differentiated cancer cells that can induce tumor formation and differentiation into various cell types within the tumor. Chronic inflammation, that is, secreted factors from stromal and immune cells, triggers signaling changes in stem/progenitor cells that abnormally activate stem cell-related molecular pathways and promote the transformation of non-malignant cells into CSCs, ultimately inducing tumor formation. The IL-6/JAK/STAT3 signaling axis plays a central role in regulating the generation of CSC in human breast cancer cell lines. Compared with other tumor cell types, IL-6/JAK2/STAT3 pathway is preferentially active in CD44+CD24- breast cancer cells, and JAK2 inhibition reduces their number and blocks the growth of xenograft (*Marotta et al., 2011*). Furthermore, IL-6 regulates the expression of CSC-related Oct-4 genes in non-CSC through the IL-6/JAK1/STAT3 signal transduction pathway (*Kim et al., 2013*). TNF-$\alpha$ induced inflammation promotes tumorigenesis and cancer progression and is a key signal for the generation of CSCs. For example, TNF-$\alpha$ triggers chromosomal instability in hepatocellular progenitors by modulating ubiquitin D and checkpoint kinase 2 and enhances the self-renewal of hepatocellular progenitors through the TNFR1/Src/STAT3 pathway, which collaboratively promotes the transformation of hepatocellular progenitors into hepatocellular carcinoma stem cells (*Li et al., 2017*). Additionally, TNF-$\alpha$ induces malignant transformation of ISCs by activating NF-$\kappa$B and Wnt/$\beta$-catenin pathways (*Zhao et al., 2020*). Overall, the cellular and molecular mechanisms of inflammatory induction of CSC generation have been gradually elucidated, providing a new molecular classification for the individualized treatment of cancer.

**Table 2  Inflammatory soluble factors and their related signals promote malignant phenotypes of cancer stem cells.**

| Soluble factors and Inflammation related signal transduction pathways | Cancer stem cell type | Effects on cancer stem cell function | References |
|---|---|---|---|
| IL-6/HIF1 | Breast cancer stem cell | Stemnness markers, sphere formation, self-renewal, metastases | *Balamurugan et al. (2019)* |
| SRSF1/circATP5B/miR-185-5p/HOXB5-IL-6/JAK2/STAT3 | Glioma stem cells | Proliferation | *Zhao et al. (2021)* |
| TNF-$\alpha$/NF-$\kappa$B | Glioma stem cells | Cell viabilities, proliferation, invasion, neurospheres formation abilities, stemness markers | *Jiang et al. (2022)* |
| PGE2/NF-$\kappa$B | Colorectal cancer stem cell | Expansion and Metastasis | *Wang et al. (2015a)* |
| CCL16/CCR2/GSK3$\beta$/$\beta$-catenin/OCT4 | Breast cancer stem cell | Stemness markers | *Shen et al. (2021)* |
| IL-17/NF-$\kappa$B/p38MAPK | Ovarian cancer stem cell | Self-renewal | *Xiang et al. (2015)* |

**Inflammation maintains the malignant phenotype of cancer stem cells**

The complex interaction between CSCs and their microenvironment can further regulate the growth of CSCs. Inflammatory cytokines, such as ILs, TNF, and chemokines, maintain the stemness state of CSCs in a variety of ways (Table 2). IL-6-mediated signal transduction has been extensively studied. For example, the transcription factor CCAA T/enhancer binding protein delta (C/EBP-$\delta$) amplifies IL-6 and hypoxia-inducible factor 1 (HIF-1) signaling by directly targeting the IL-6 receptor (IL6RA), whose deletion or depletion reduces the expression of stem cell factors and stem cell markers, the formation of blobs, and self-renewal (*Balamurugan et al., 2019*). The expression of circATP5B in glioma stem cells (GSC) was significantly up-regulated and promoted the proliferation of GSC. Mechanically, circATP5B acts as a miR-185-5p sponge to up-regulate the expression of homeobox protein Hox-B5 (HOXB5). HOXB5 is overexpressed in gliomas and transcriptionally regulates IL6 expression (*Zhao et al., 2021*). IL-6 also promotes the cell stemness and invasiveness of glioblastoma by inhibiting the expression of miR-142-3p (*Chiou et al., 2013*). Alternatively, TNF-$\alpha$-mediated signaling pathways have also been highlighted. Long-term exposure to TNF-$\alpha$ can increase the proportion of CSCs in oral squamous cell carcinoma, thereby enhancing its pellet-forming ability, stem cell transcription factor expression, and tumorigenicity (*Lee et al., 2012*). CircKPNB1 overexpression can promote the proliferation, invasion, and stem cell viability of GSC. Mechanistically, circKPNB1 regulates protein stability and nuclear translocation in SPI1. SPI1 promotes the malignant phenotype of GSC through TNF-$\alpha$ mediated NF-$\kappa$B signaling (*Jiang et al., 2022*). Other inflammatory factors, such as PGE2, chemokine (C-C motif) ligand (CCL) 16, and IL-17(*Wang et al., 2015a*; *Shen et al., 2021*; *Xiang et al., 2015*). can promote the malignant phenotype of CSCs through relevant signaling pathways. Overall, the evidence suggests that inflammatory cytokines are involved in the mechanism of CSCs stemness, and targeting inflammatory cytokines may be a useful adjunct to cancer therapy.

## Influence of inflammation on the spread of cancer

The importance of intracellular communication between cancer and immune cells in the tumor microenvironment has long been recognized (*Stoeltzing, Meric-Bernstam & Ellis, 2006*). When stimulated, host immune cells secrete cytokines and other tiny inflammatory proteins to fight cancer; however, these cytokines sometimes activate cancer cells and lead to specific mutations and epigenetic changes (*Galdiero & Mantovani, 2018*). In most cases, chronic inflammation is required for the development of cancer (*Coussens & Werb, 2002*; *Anuja et al., 2017*). Cases of chronic inflammation resulting from tissue damage leading to cancer have lost their clinical novelty. For example, the tobacco carcinogen nicotine-derived nitrosamine ketone significantly promotes lung cancer by increasing the expression of CCL20 (*Wang et al., 2015b*). Obesity also induces chronic inflammation (activation of IL-6 and TNF-$\alpha$) and promotes hepatocellular carcinoma (*Park et al., 2010*). Helicobacter pylori infection is associated with gastric cancer and gastric mucosa-associated lymphoid tissue lymphoma (*Fischbach & Malfertheiner, 2018*). Human papillomavirus infection leads to lesions of the cervical squamous epithelium and progression to cervical cancer (*The Cancer Genome Atlas Research Network, 2017*). Although an immune response to pathogens is a natural defense, all these pathogens can cause long-term infections, which can induce chronic inflammation and, ultimately, cancer cell formation. These examples demonstrate that almost all forms of cancer, regardless of carcinogenic factors, are associated with immune activation and chronic inflammation. Therefore, it is important to explore the influence of the inflammatory microenvironment on the behavior of cancer cells.

### Inflammatory microenvironment reprograms cancer cell fate

Cytokines and growth factors produced during chronic inflammation may have multifunctional effects on tumor formation and growth, both directly on tumor cells and indirectly by promoting favorable conditions in the microenvironment. DNA damage is a bridge between chronic inflammation and cancer (*Punt et al., 2016*). During inflammation, reactive oxygen and nitrogen species are created to fight pathogens and stimulate tissue repair and regeneration; however, they also damage DNA, which in turn promotes mutations that lead to genomic instability that can initiate and promote cancer development (*Kay et al., 2019*; *Colotta et al., 2009*; *Pikor et al., 2013*). The ability of inflammation to reprogram cancer cell fate is linked to cancer development. Persistent inflammation promotes cancer development by activating the proliferation, survival, and metastasis of cancer cells. (*Pikarsky et al., 2004*; *Kiraly et al., 2015*) For example, CCL2-CCR2 signaling promotes cancer progression by supporting cancer cell proliferation and survival, inducing cancer cell migration and invasion, and stimulating inflammation and angiogenesis (*Xu et al., 2021*). IL-6 directly affects tumor growth by enhancing the proliferation and survival of malignant cells in multiple myeloma, non-Hodgkin's lymphoma, and hepatocellular carcinoma (*Aggarwal et al., 2006*; *He & Karin, 2011*). It has been found that TNF-$\alpha$ and IL-1$\beta$ can activate the hypoxia signaling pathway in human liver cancer cells, regulate tumor cells, and directly affect tumor growth (*Hellwig-Bürgel et al., 1999*). Chronic inflammation caused by smoking can cause neutrophils to exhibit suicidal anti-infective behavior, which, in addition to awakening cancer cells that are dormant for years or even decades in patients

with cancer, causes tumor recurrence (*Albrengues et al., 2018*). Conversely, anticancer compounds inhibit cancer initiation and progression by downregulating pro-inflammatory cytokine levels (*Keshari et al., 2017*; *Barker et al., 2018*; *Chikara et al., 2018*).

### Inflammatory microenvironment promotes cancer cell metastasis

Epithelial-to-mesenchymal transition (EMT) is an embryonic process that loosens cell–cell adhesion complexes and confers enhanced migratory and invasive properties to cells, and it is exploited by cancer cells during metastasis. Cancer cells that undergo EMT are more aggressive and exhibit enhanced invasiveness, stem cell-like characteristics, and anti-apoptotic capabilities. Inflammation is a potent inducer of EMT in tumors; conversely, EMT can also stimulate cancer cells to produce pro-inflammatory factors (*Suarez-Carmona et al., 2017*). Researchers have comprehensively summarized the progression of inflammation and EMT in cancer cells (*Suarez-Carmona et al., 2017*). Briefly, the EMT/inflammatory axis promotes the aggressiveness of primary tumors, and EMT and inflammatory markers are associated with poor prognosis in multiple cancer patient cohorts. In addition to acting as a direct inducer of EMT in cancer cells, inflammation can act as an intermediate mediator of cancer development. For example, microenvironment dysregulation due to microbial interactions promotes tumor development. Microbes do not directly influence tumor behavior and require mediators to promote tumor development. The current view is that microorganisms regulate tumor immunity through their derived metabolites, toxins, antigens, and other substances; they regulate tumor cell metabolism and reshape the tumor microenvironment to promote tumor occurrence and progression. Inflammation acts as an intermediary in this process (*Sepich-Poore et al., 2021*). In addition, the regulatory balance between tumors and immunity is a cyclic process. For example, one study found that tumor cells secreted granulocyte-macrophage colony-stimulating factor to stimulate macrophages, which were activated and secreted CCL18, promoting tumor cell EMT and eventually leading to lung metastasis of breast cancer cells (*Su et al., 2014*). Altogether, the main cause of cancer death is not the primary tumor itself but the depletion of distant organs and tissue metastases. Although the mechanism underlying this process is unclear, inflammation-mediated EMT plays an important role.

In general, malignant tumors are difficult to treat because, in addition to the ability of cancer cells to proliferate, invade, and metastasize indefinitely, they can evade immune surveillance. A series of immunotherapeutic approaches based on immune evasion mechanisms have been developed and clinically applied in the past few decades. Unlike traditional chemoradiotherapy, immunotherapy mainly uses immune cells inside and outside the tumor microenvironment to identify and attack cancer cells (*Yost et al., 2019*), which theoretically makes immunotherapy more specific with few side effects. Nonetheless, downregulation of major histocompatibility complex class I antigen presentation, which frequently occurs in solid cancers, limits the effectiveness of these therapies. Research has confirmed that cells that appear to be in the quiescent phase are resistant to immune system attacks (*Bruschini, Ciliberto & Mancini, 2020*). The ability of long-lived stem cells to evade immune surveillance may be due to their mostly non-proliferating quiescent state, which may be an important feature of stem cells that develop into cancer.

## CONCLUSIONS

The inactive division of CSCs is key to cancer metastasis, recurrence and drug resistance. Although compounds and methods have been found to specifically inhibit or eliminate CSCs (*Shi et al., 2019*; *Sato et al., 2021*; *Ohta et al., 2022*). no systematic studies have been conducted to achieve the eradication of CSCs *in vivo*. While the origin of CSCs can vary by tissue evidence suggests that ASCs can transform into CSCs under continuous abnormal activation of pathways related to cell proliferation and survival. This abnormal activation is typically triggered by the changes in stem cell niche, and inflammation is one of the key factors that can mediate such changes. ASCs are usually present in a quiescent state and function to maintain tissue homeostasis. During tissue injury, ASCs exit the quiescent state, driven by complex signals including inflammation, and enter the cell cycle to repair the damaged tissue through division and differentiation. However, prolonged periods of ASCs division increase the risk of cancer (Fig. 3). This partly explains why ISCs, skin stem cells, and stomach epithelial cells renew more quickly than other cells in the body, and they are at a high risk of developing cancer (*Sánchez-Danés et al., 2016*; *Hayakawa et al., 2021*; *Aliluev et al., 2021*). Conversely, non-regenerating cells such as nerve cells and cardiomyocytes have a significantly low risk of developing cancer at their tissue sites.

Although the concept of inflammation-inducing cancers has gained acceptance over time, its underlying mechanism has not been clearly explained. Based on the stem cell division theory of cancer, this review highlights the impact of inflammation on normal stem cells, CSCs, and cancer cells. It also speculates on the potential of inflammation to promote the transformation of ASCs into CSCs and to facilitate cancer metastasis. Ultimately, a comprehensive understanding of how inflammation influences the behavior of stem cells and cancer cells could revolutionize our comprehension of numerous diseases, paving the way for the development of novel therapeutic interventions. For example, chemotherapy kills only proliferating cancer cells, but the survival of quiescent cancer cells, or cancer stem cells (non-dividing), is crucial for cancer to return. The tumor microenvironment, especially changes in the composition and number of inflammatory cells or cytokines, can activate cancer cells or CSCs to divide. Therefore, the appropriate combination of inflammation and chemotherapy may result in a more effective cancer treatment.

### Abbreviations

| | |
|---|---|
| **AP-1** | activating protein 1 |
| **ASCs** | adult stem cells |
| **ATM** | ataxia telangiectasia-mutated gene |
| **ATR** | ATM and rad3-related gene |
| **BST2** | bone marrow stromal cell antigen 2 |
| **CCL** | chemokine (C-C motif) ligand |
| **C/EBP-$\delta$** | CCAA T/enhancer binding protein delta |
| **cGAS** | cyclic GMP-AMP synthase |
| **CLCs** | clara-like cells |
| **CLRs** | C-type lectin receptors |
| **CRC** | colorectal cancer |

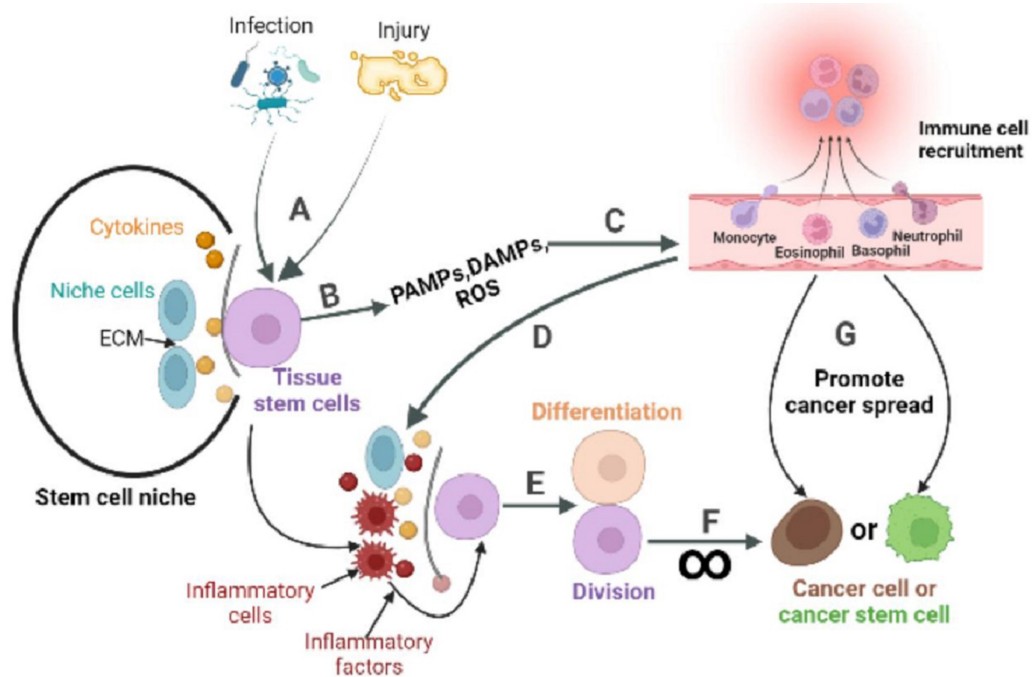

**Figure 3** **The role of inflammatory response in the stem cell division theory of cancer.** Tissue stem cells are quiescent in the niche. (A) The organism is infected with a pathogen or has endogenous tissue damage. (B) Tissue stem cells produce PAMPs, DAMPs, and ROS upon stimulation by pathogens or endogenous danger signals. (C) PAMPs and DAMPs activate pattern recognition receptors (PRR). The PRR signaling pathway is the promoter of a cascade of reactions that ultimately lead to the migration of immune cells to infection site. The innate immune response is subsequently activated, while the adaptive immune response can be activated directly or indirectly. (D) Recruited immune cells alter the tissue stem cell niche, causing inflammatory cells and factors to become its major components. (E) Inflammatory cells or factors act directly or indirectly on tissue stem cells to activate their regenerative capacity. (F) In response to repeated inflammatory stimulation, tissue stem cells undergo long-term division, resulting in the accumulation of genomic mutations, and eventually mutate into cancer cells. (G) Inflammation also reprograms the fate of cancer cells, or cancer stem cells, and promotes the spread of cancer.

| | |
|---|---|
| **CSCs** | cancer stem cells |
| **DAMPs** | damage-associated molecular patterns |
| **EMT** | epithelial-to-mesenchymal transition |
| **EP** | prostaglandin E receptor |
| **G-MSCs** | gingival mesenchymal stem/progenitor cells |
| **gp130** | glycoprotein 130 |
| **GSC** | glioma stem cells |
| **HIF-1$\alpha$** | hypoxia-inducible factor 1 alpha |
| **HOXB5** | homeobox protein Hox-B5 |
| **HSC** | hematopoietic stem cell |
| **IFN** | interferon |
| **IL** | Interleukin |
| **IKK** | I$\kappa$B kinases |
| **ISCs** | intestinal stem cells |

| | |
|---|---|
| **Jag1** | Jagged 1 |
| **JAK** | janus kinases |
| **JNKs** | jun N-terminal kinases |
| **LPS** | lipopolysaccharides |
| **MAPK** | mitogen-activated protein kinase |
| **MSCs** | mesenchymal stem cells |
| **MuSCs** | muscle stem cells |
| **MYD88** | myeloid differentiation primary response gene 88 |
| **NF-$\kappa$B** | nuclear factor-k-gene binding |
| **NLRs** | Nod-like receptors |
| **Nrf2** | Nuclear factor erythroid2-related factor 2 |
| **PAMPs** | pathogen-associated molecular patterns |
| **PGE2** | prostaglandin E2 |
| **PRRs** | pattern recognition receptors |
| **RLRs** | RIG-i-like receptors |
| **ROS** | reactive oxygen species |
| **STAT** | signal transducer and activator of transcription |
| **STING** | stimulator of interferon genes |
| **Th** | T helper |
| **TIR** | Toll–IL-1-resistence |
| **TLRs** | Toll-like receptors |
| **TNF** | tumor necrosis factor |
| **Treg** | regulatory T cells |
| **TRIF** | toll-receptor-associated activator of interferon. |

### Funding

This work was supported by the National Natural Science Foundation of China, PR China (grant numbers: 31960191; 82260158), Science and Technology Innovation Leading Academics of National High-level Personnel of Special Support Program, Ministry of Science and Technology, PR China (grant number: GKFZ-2018-29), Guizhou High-Level Innovative Talent Support Program, PR China (grant number: QKHPT-RC-GCC[2022]001-1), and S&T Foundation of Guizhou, PR China (grant number: QKHQKHJC-ZK-2021-ZD-026). The funders had no role in study design, data collection and analysis, decision to publish, or preparation of the manuscript.

### Grant Disclosures

The following grant information was disclosed by the authors:
National Natural Science Foundation of China, PR China: 31960191, 82260158.
Science and Technology Innovation Leading Academics of National High-level Personnel of Special Support Program.
Ministry of Science and Technology, PR China: GKFZ-2018-29.
Guizhou High-Level Innovative Talent Support Program, PR China: QKHPT-RC-GCC[2022]001-1.
S&T Foundation of Guizhou, PR China: QKHQKHJC-ZK-2021-ZD-026.

## Competing Interests

The authors declare there are no competing interests.

## Author Contributions

- Yi Luo performed the experiments, analyzed the data, prepared figures and/or tables, authored or reviewed drafts of the article, and approved the final draft.
- Jian-Hui Xiao conceived and designed the experiments, authored or reviewed drafts of the article, and approved the final draft.

## Data Availability

 This is a literature review.

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
