# Peer review of "Inflammatory auxo-action in the stem cell division theory of cancer"

_PeerJ, doi:10.7717/peerj.15444_

## Round 0.1 · original submission · Major Revisions

Your PeerJ submission entitled " The role of inflammatory responses in stem cell division theory of cancer" has been reviewed, including by a member of our editorial staff who has commented on the overall organization and presentation of your review. The reviewers were of the opinion that the manuscript contains important information of interest to other investigators. However, they also identified several major concerns that require further attention, as indicated in the reviews appended below.

Some of my critiques are as below:

- The abstract needs to be rewritten as there is a missing link between the introduction part and the conclusions, in other words, the rationale of the review needs to be clear.

- The review is aimed to demarcate the influence of inflammatory responses on cancer stem cell transition and progression, involvement of both positive and negative effects of inflammation on cancer stem cells needs to be covered.

- I see several repeated sentences which need to be corrected throughout the manuscript. Subtitles 4 and 5 are quite similar and need to be justified.

- There are several undefined abbreviations throughout the manuscript, they need to be explained least at the first instance.

Reviewer 1 ·

Basic reporting

In this review, the authors focused on the stem cell division theory of cancer and summarized the influence of the inflammatory microenvironment on cancer occurrence and development from the perspective of stem cells and cancer cells.
The topic has not been reviewed extensively recently. Also, the author provided a lot of background information about the oncogenesis processes and other related fields.
The language used in the review is clear, unambiguous and professional.
The structure of the review is well-arranged. However, the subtitle of part 4 (line 378) and 5 (line 431) are extremely similar which can cause confusion to the readers.
It would be better if the author can put more detailed description for figure 1.
Also, the authors mentioned the differences between acute and chronic inflammation in the context of cancer immunology. It would be better if such information can be shown in figure 2.
There are some small mistakes in the manuscript.
Line 264, "CCl4-induced" should be "CCL4-induced"

Experimental design

The survey methodology is consistent with a comprehensive, unbiased coverage of the subject according to the authors in the survey methodology part.
The sources are adequately cited. Also the review is organized into coherent paragraphs and subsections.

Validity of the findings

The information the authors reviewed will be of interest to researchers working in the related field.

Reviewer 2 ·

Basic reporting

The manuscript is well organized and written coherently. This review aims to delineate the influence of inflammatory signals and their pathways on cancer stem cell transition and progression, which ultimately leads to tumor formation. The manuscript sufficiently introduced non-malignant stem cells, cancer stem cells, and inflammatory pathways influence. The topic itself is extensive and relevant to current trends in the cancer development field. Even though the effect of inflammatory signals on a single type of cancer was reviewed elsewhere, this manuscript reviewed inflammatory pathways in different types of cancers.

Experimental design

The manuscript rigorously reviewed the literature to prove the importance of inflammatory signals in their contribution to cancer development. The authors cleverly divided the manuscript into three subsections. In the first section, the influence of inflammation on stem cell behavior, dormant stem cell fate, and regeneration in response to injury was explained. In the following two sections, the manuscript describes the role of inflammatory pathways on cancer stem cells and cancer cells in terms of their generation, maintenance, and behavior.
• The main drawback of the manuscript, there are several repetitive sentences. For example, sentences in lanes 182 and 275 explained the quiescent cell phenotype. Please removes these repetitive sentences that can be found throughout the manuscript.
• The introduction did not explain the other confounding factors resulting in cancer development. The authors jumped straight to the role of inflammatory pathways in cancer development. Please include a paragraph on the factors leading to cancer incidence and focus on the role of the inflammatory pathway.
• Both tables need to be reformatted for better readability. Please includes columns.

Validity of the findings

The arguments presented in the manuscript very well supported the aim of the study that inflammation plays a significant role not only in stem cell injury and repair. Further, arguments supported the influence of inflammation signals in cancer development through cancer stem cell generation. However, the manuscript failed to introduce the role of cell protective pathways such as senescence to prevent inflammation signals leading to cancer development. Overall, the conclusions made by the authors were proven to be correct and supported by the literature review.

·

Basic reporting

Comments:
1. Authors need to revise the abstract. I do not understand, how inflammation form the cancer cells. There is redundancy so authors have to rewrite the abstract.
2. In the introduction section line 95-116 authors discussed the correlation between DNA alterations and cell division and how these alterations accumulate the mutational burden in normal as well as in stem cells. However, it will be more informative if authors can discuss how the different DNA errors affect the inflammatory responses in stem cells that convert them into cancer stem cells.
3. Line 102-105 “A landmark research has demonstrated that cancer incidence shows a strong correlation with the accumulation of stem cell divisions in tissues; variation in cancer risk among tissues can be explained by the lifetime number of tissue-specific stem cell divisions.” I do not understand what the authors want to say here.
4. Since the author's review is focused on the effect of inflammatory responses on stem cells and cancer progression but I feel in the background/ introduction is very weak authors didn’t put much information. Authors need to provide focus background information about this part. It will be easy for the readers to understand the purpose of the review.
5. Line 203-208 “Additionally, chronic inflammation contributes to the establishment of an immunosuppressive tumor microenvironment by recruiting a variety of immunosuppressive cells including immunosuppressive cells including tumor-associated M2-macrophages, myeloid-derived suppressor cells, and regulatory T (Treg) cells, thereby promoting tumorigenesis and progression. Therefore, chronic inflammation has been proven to be one of the causes for the occurrence and development of cancer”. Authors need to check this para and explain briefly what type of specific immunosuppressive cells. Also, cite some relevant papers which support this statement.
6. Line 209-214 “Inflammation is an adaptive response triggered by infection and tissue damage and is differentiated by pathogen-associated molecular patterns (PAMPs) and damage-associated molecular patterns (DAMPs). PAMPs, such as membrane-associated lipids and lipopolysaccharides, are exogenous components that are specific to invading microorganisms. In contrast, inflammation without pathogens and their products is called sterile inflammation and is triggered by endogenous danger signs”. Authors should cite some relevant papers. I am wondering why the authors didn’t provide more information about the DAMs and PAMPs-mediated inflammatory signaling pathways. Authors need to look at these relevant papers.
a. Rossi, Fiorella, Hunter Noren, Richard Jove, Vladimir Beljanski, and Karl-Henrik Grinnemo. "Differences and similarities between cancer and somatic stem cells: therapeutic implications." Stem Cell Research & Therapy 11, no. 1 (2020): 1-16.
b. Locy, Hanne, Sven De Mey, Wout De Mey, Mark De Ridder, Kris Thielemans, and Sarah K. Maenhout. "Immunomodulation of the tumor microenvironment: turn a foe into a friend." Frontiers in immunology 9 (2018): 2909.
c. Chefetz, Ilana, Ayesha Alvero, Jennie Holmberg, Noah Lebowitz, Vinicius Craveiro, Yang Yang-Hartwich, Gang Yin et al. "TLR2 enhances ovarian cancer stem cell self-renewal and promotes tumor repair and recurrence." Cell Cycle 12, no. 3 (2013): 511-521.
Lines 230-240 “Influence of inflammation on nonmalignant stem cell behavior” I am struggling to find any information about this and what authors want to discuss in this section. How does inflammatory signaling affect the normal cells
8. In line 241; “Tissue stem cells, which reside in various tissues and organs, are generally dormant.” Are these stem cells normal stem cell that requires tissue homeostasis or dormant? I am a bit confused. It would be clearer if the authors can explain the difference between normal and dormant stem cells.
9. In Lines 383-385 “Many signaling pathways that contribute to the survival, proliferation, self-renewal, and differentiation characteristics of nonmalignant stem cells are abnormally activated or inhibited in CSC.”
I am curious to understand if authors can discuss the pathways that are up and downregulated and responsible for cell survival and proliferation etc.
10. In lines 386-387 What are the necessary pathways for inhibition to activate the oncogenic factors in normal stem cells to convert CSCs? Can authors provide some data?
11. In lines 471-494 “Inflammatory microenvironment promotes cancer cell metastasis” I am not convinced with the evidence provided by the authors. GMCSF is not the only factor that induces EMT. There are certain other proinflammatory mediators, such as cytokines, chemokines, and matrix metalloproteinases (MMPs), which fuel the cancer‐related inflammation and certain activated resident cells, such as endothelial cells or CAFs or resident macrophages (TAMs) or dendritic cells including the tumor milieu recruits bone marrow‐derived cells, mostly neutrophils, macrophages, and immature, immunosuppressive myeloid cells called myeloid‐derived suppressive cells (MDSCs). I would strongly recommend that authors need focus on these aspects also.
12. The whole conclusion part is very weak. At most the point, the authors fail to conclude and also fail to compare with the published evidence which supports their review. I would strongly recommend that the authors need to rewrite the conclusion part.
13. In general authors used several abbreviations. I would recommend that authors need to add the full form initially to avoid confusion.
14. In a few instances, the spaces between adjacent words are missing. There are also grammatical mistakes. Please check the spelling and grammar. I would suggest that the whole manuscript be thoroughly revised to improve clarity and readability.
15. Reference section needs revisions and authors need to cite more recent papers in their whole manuscripts.

Experimental design

No comment

Validity of the findings

No Comment

Reviewer 4 ·

Basic reporting

The authors have reviewed the link between inflammation and its influence on cancer stem cells and cancer progression. Many studies have indicated both positive and negative effects of inflammation on cancer stem cells. Both aspects need to be discussed.

Major revisions-
1. The rationale to write the current review in unclear. If authors include recent developments about inflammatory regulators in the tumor microenvironment in the abstract section, it will enhance the readability.
2. In the introduction section, key findings about the inflammatory responses in modulation of cancer stem cell behavior must be included and then expanded in the subsequent sections.
3. Separate section about different immune cells in regulation of tumor microenvironment must be included.
4. There are many reports that suggest inflammation lowers cancer growth. Such studies needs to be incorporated.
5. Future studies and open questions must be included to stimulate further research.

Experimental design

No comments

Validity of the findings

Not applicable to review

Additional comments

None

---

## Round 0.2 · accepted · Accept

The reviewers are satisfied with the clarification given and the modified version of the manuscript. The authors have addressed most of the comments by the reviewers.

Reviewer 1 ·

Basic reporting

The authors have answered the points properly.

Experimental design

The authors have answered the points properly.

Validity of the findings

The authors have answered the points properly.

·

Basic reporting

My all comments has been addressed. However, I would still suggest if authors can reframe the first para of the introduction.

Experimental design

My all comments has been addressed.

Validity of the findings

My all comments has been addressed.

Additional comments

My all comments has been addressed.

Reviewer 4 ·

Basic reporting

Clear and easy to follow.
Timely review.

Experimental design

Review organized logically.

Validity of the findings

Most of the recent works were cited.

Additional comments

Accepted in the present format.